

# Stance detection with BERT embeddings for credibility analysis of information on social media

Hema Karande[1], Rahee Walambe[2,3], Victor Benjamin[4], Ketan Kotecha[1,2] and TS Raghu[4]

[1] Computer Science, Symbiosis Institute of Technology, Pune, Maharashtra, India
[2] Symbiosis Centre for Applied Artificial Intelligence, Symbiosis International (Deemed University), Pune, India., Pune, Maharashtra, India
[3] Electronics and Telecommunication, Symbiosis Institute of Technology, Pune, Maharashtra, India
[4] Department of Information Systems, Arizona State University, Phoenix, AZ, United States of America

Corresponding authors
Rahee Walambe,
rahee.walambe@scaai.siu.edu.in
Ketan Kotecha, head@scaai.siu.edu.in

## ABSTRACT

The evolution of electronic media is a mixed blessing. Due to the easy access, low cost, and faster reach of the information, people search out and devour news from online social networks. In contrast, the increasing acceptance of social media reporting leads to the spread of fake news. This is a minacious problem that causes disputes and endangers the societal stability and harmony. Fake news spread has gained attention from researchers due to its vicious nature. proliferation of misinformation in all media, from the internet to cable news, paid advertising and local news outlets, has made it essential for people to identify the misinformation and sort through the facts. Researchers are trying to analyze the credibility of information and curtail false information on such platforms. Credibility is the believability of the piece of information at hand. Analyzing the credibility of fake news is challenging due to the intent of its creation and the polychromatic nature of the news. In this work, we propose a model for detecting fake news. Our method investigates the content of the news at the early stage i.e., when the news is published but is yet to be disseminated through social media. Our work interprets the content with automatic feature extraction and the relevance of the text pieces. In summary, we introduce stance as one of the features along with the content of the article and employ the pre-trained contextualized word embeddings BERT to obtain the state-of-art results for fake news detection. The experiment conducted on the real-world dataset indicates that our model outperforms the previous work and enables fake news detection with an accuracy of 95.32%.

## INTRODUCTION

In the information age, social networking sites have become a hotbed for spreading misinformation. Misinformation (*Soll, 2016*) as a phenomenon is as old as true or factual ones. The scale and scope of misinformation, however, have assumed alarming levels as social media platforms and networks can spread misinformation rapidly. With the

substantial increase in the spread of misinformation, adverse impacts on individuals and society at large have also become significant (*Figueira & Oliveira, 2017*). In this work, we propose a framework for identifying the misinformation by employing state-of-the-art artificial intelligence algorithms. The first step in the identification of misinformation is to understand what constitutes the misinformation. Fake news, misinformation, disinformation all are various forms of non-factual information with variations in the intent of the creator/spreader. Ethical Journalism Network (https://ethicaljournalismnetwork.org/tag/fake-news/page/2 (EJN) defines fake news as "information deliberately fabricated and published to deceive and mislead others into believing falsehoods or doubting verifiable facts." Misinformation, disinformation, and mal-information are specific subsets of information disorder. Disinformation is false and is deliberately designed to harm an individual, organization, social group, or country. Mal-information is reality-based, used to harm a person, social group, organization, or country (UNESCO: https://en.unesco.org/fightfakenews). Misinformation is not created to cause harm and is false information that is disseminated by people who believe that it is true. In comparison, disinformation is false information intentionally and knowingly propagated by individuals to gain political, financial, social, or psychological benefits. Disinformation via advertising can be used to discredit or falsely credit a product or a specific manufacturer for financial gain (Bannerflow: https://www.bannerflow.com/blog/marketing-fake-news-dangerous-game/). In the political domain, disinformation could manifest from using false information to discredit opposition parties or creating false narratives to aid one specific party or candidate (*Allcott & Gentzkow, 2017*). Socially, one typical example could be the spread of certain medical myths that are prevalent in specific communities and spreading them without much thought (*Waszak, Kasprzycka-Waszak & Kubanek, 2018*).

Misinformation or Information Disorder is usually in the form of false or out of context information, photographs, or videos that are either intentionally created and distributed. Sometimes, they are taken out of context to mislead, deceive, confuse or misinform the reader (*Pérez-Rosas et al., 2017*). Although there is news created for fun and circulated as a part of a joke they have seriously impacted society. Researchers (*Friggeri et al., 2014*) surveyed different aspects of false information and answered the question "what can be termed as false?" The primary points considered are who is spreading the false information, what are the reasons behind the reader's belief, and what is impact this false news creates. The effects of dis /misinformation on society can prove detrimental. Misinformation has caused a serious impact on various activities such as affecting the stock market (*Bollen, Mao & Zeng, 2011*), hampering the responses during natural disasters (*Gupta et al., 2013*), instigating terrorism activity (*Starbird et al., 2014*), kindling cyber-troop activity (*Bradshaw & Howard, 2018*), hampering the decision-making ability during elections (*News18, 2019*) and creating panic, bringing about the economic crisis (*Herald D, 2020*) and inciting religion-based attacks (*Indianexpress,* ) during the COVID-19 pandemic.

Looking at the huge outburst of fake news around the coronavirus, the World Health Organization (*World Health Organization, 2020*) announced the new coronavirus pandemic was accompanied by a 'Misinformation Infodemic'. Various aspects of misinformation and its identification using AI tools for COVID 19 data is reported in a

recent article (*Jyoti Choudrie , 2020*). Fact Checkers report fake news from general elections and attacks at Pulwama to the scrapping of Article 370 and the ongoing protests against the Citizenship Amendment Act, which triggered a wide distribution of misinformation across social media platforms (*Chaturvedi, 2019*). Misinformation affects communities and their susceptibility in various ways; for instance, mob lynching and communal poison.

The dependability of mass on social media news items has rapidly grown. It is reported that out of the English-speaking news seekers in India 84 percent rely on Online news whereas 16 percent on the outpaced print media (*Aneez et al., 2019*). The urban, semi-urban teen-agers are the primary consumers of social media news (*Pérez-Rosas et al., 2017*). Due to such tremendous use of online platforms, the spread of information disorder is overwhelming and immense, causing harm to society. In the year 2014, the World Economic Forum declared misinformation as one of the 10 global risks (*W.E.Forum, 2014*). Governments have taken some anti-misinformation efforts aiming to curb the spread of unlawful content and misinformation spanning from the laws, Media literacy campaigns, government task force, bills, platform agreements, to arrests (*Funke & Flamini, 2019*).

From the social media platforms available, Facebook and WhatsApp are particularly widely used for news updates. As reported by Reuters, 75% use Facebook, 82% use WhatsApp, 26% use Instagram, 18% use Twitter. Hence it becomes the responsibility of these platforms to help mitigate the spread of misinformation. Facebook (https://www.facebook.com/facebookmedia/blog/working-to-stop-misinformation-and-false-news) states that they are working in different ways –e.g., most false news is motivated due to financial aspects, hence undermining the economic incentives may prove useful. The International Fact-Checking Network and the fact-checkers are working hard to investigate the facts behind a piece of information likely to be fake. Several experiments were carried out to assess the effect of hoaxes, false reviews, and fake news. To create a misinformation detection system, we need to consider the various aspects of the knowledge and categorization of different features. Several researchers performed research and submitted it. We present the literature in parts that concentrate on social and cognitive dimensions, categorization strategies, and AI-based detection systems using different acceptable algorithms.

Mis- and disinformation can easily be disseminated—wittingly or unwittingly—through all types of media. The ease of access to such quick information without any validation, has put a responsibility on the reader to decide the correctness of the information at hand. Correctness, trustworthiness, or credibility are the qualities of the information to be trusted and believed in. In the context of news, it encompasses the broader aspects of trustworthiness, reliability, dependability, integrity, and reputation. When people are unable to debunk the information and act accordingly, that makes poor decisions impacting their lives. It is essential to check the credibility of the source, author, check your biases, check the date and supporting sources to determine the reliability via comparison with reliable sources (EMIC: https://guides.emich.edu/newseval).

While performing the credibility analysis we need to first examine how misinformation and disinformation are being represented, spread understood, and acted upon. The role

and motivation of an individual behind resharing the original content is an important aspect while devising a policy to curtail the spread and also for developing some technical solutions to mitigate it.

The most powerful of the information disorder content is that which harms people by influencing their emotions. Since the social platforms are designed to express emotions through likes, comments, and shares, all the efforts towards fact-checking and debunking false information are ineffective since the emotional aspect of the sharing of information is impossible to control. Detection and mitigation of information disorder are challenging due to the psychological aspects of the motivation for dissemination and the proliferation of misinformation. The two primary channels for spreading the misinformation are employed namely, Echo Chamber (*Shu, Bernard & Liu, 2019*) which is a situation where beliefs are reinforced or enhanced by contact and repetition within a closed structure and Filter Bubble (*Shu, Bernard & Liu, 2019*) is the state of intellectual isolation that can result from custom searches when a website algorithm selectively estimates what information a user wants to see based on user information, such as location, past click history, and search. The concept of the filter bubble is used to target a specific group of people to spread the specific misinformation. As per *Kumar & Geethakumari (2014)* cognitive psychology plays an important role in the spread of misinformation.

As stated earlier, there are political, financial, and social aspects that play a role as a motivation behind the creation of fake news items. These diverse angles, namely, the dynamic and ubiquitous nature of information, difficulty in verification, and homophily prove to be some of the primary challenges in finding the credibility of the information.

## Previous work

Misinformation detection is studied in different ways, starting with how it is created, spread, and eventually affects the community. *Shu et al. (2017)* surveys the literature from two distinct phases: characterization and detection. Characterization is concerned with understanding the basic concepts and principles of fake news in traditional and social media whereas data mining with feature extraction and model construction is included in detection. *Shu et al. (2017)* presents the characteristics of Fake News on traditional and social media that include Psychological and social foundations as well as fake accounts and echo chamber creation on social media. The author also puts forward the detection approaches including News Content and Social Context.

Various approaches towards fake social media news identification are proposed, including data orientation, feature orientation, model orientation, and application orientation. Depending on these approaches multiple systems have developed that concentrate on temporal features, psychology, or the data for a data-oriented approach. Much explored approaches are feature orientation that considers the content or the social context of the news. Depending on the dataset the model is selected either to be supervised, unsupervised, or semi-supervised (*Shu et al., 2017*). Feature Selection is an important step while approaching fake news detection. Features are broadly categorized into content features and social context features by *Cao et al. (2018)*. The content features include

lexical, syntactic, and topic features whereas social context features include user features, propagation features, and temporal features.

There is vast work done in detecting misinformation with various approaches, traditionally some classification methods used were Decision Tree & Bayesian networks (*Castillo, Mendoza & Poblete, 2011*), Random Forest & SVM (Kwon et al., 2013), Logistic Regression (*Castillo, Mendoza & Poblete, 2011*) for the handcrafted features. The features like author, context, and writing style (*Potthast et al., 2017*) of the news can help in identifying the fake news, although writing style alone cannot be a good option. Linguistic signs may be used to identify language characteristics such as n-grams, punctuation, psycholinguistic characteristics, readability, etc. Classification based on the credibility of the person who liked it is an approach taken in some cases (*Shu et al., 2017*). The conventional techniques of machine learning have often resulted in a high-dimensional representation of linguistic information leading to the curse of dimensionality where enormous sparse matrices need to be treated. This issue can be solved with the use of word embeddings, which gives us low dimensional distributed representations. Misinformation, specifically a news item, may constitute words, sentences, paragraphs, and images. For applying any AI technique on text firstly we need to format the input data into a proper representation that can be understood by the model we are designing. Different state-of-art representation techniques like one-hot encoding, word embeddings like Continuous Bag of Words and Skip-gram (*Mikolov et al., 2013*) that compute continuous vector representations of very big datasets of terms, GloVe is Global word representation vectors (*Pennington, Socher & Manning, 2014*) global corpus statistics that train just on non-zero elements in a word-word matrix, and not on the entire sparse matrix or single background windows in a large corpus. BERT (*Devlin et al., 2018*) bi-directional pre-training setup, using the transformer encoder. Open-AI GPT pre-training model internally using the transformer decoder concept. Pre-trained embeddings can be adapted to build a neural network-based fake news detection model. Text data is a sequential time series data which has some dependencies between the previous and later part of the sentence. Recurrent Neural Networks has been widely used to solve NLP problems, traditionally encoder decoders architecture in Recurrent Neural Network was a good option where an input sequence is fed into the encoder to get the hidden representation which is further fed to the decoder and produce the output sequence. One step towards fake news detection was to detect stance (*Davis & Proctor, 2017*) that involves estimating the relative perspective (or stance) of two texts on a subject, argument, or problem. This can help in identifying the authenticity of the news article based on whether the headline agrees with, disagrees with, or is unrelated to the body of the article. Recurrent Neural Networks (*Shu et al., 2017*; *Ma et al., 2016*) to capture the variation of contextual information, CSI model composed of three modules (*Ruchansky, Seo & Liu, 2017*) implements Recurrent Neural Network for capturing user activity's temporal pattern, learning the source characteristic based on user behavior, and classifying the article. Researchers have also investigated the Rumor form of the news that is circulated without confirmation or certainty to facts (*DiFonzo & Bordia, 2007*). A rumor detection system (*Cao et al., 2018*) for Facebook (notify with a warning alert), Twitter(credibility rating is provided and the user is allowed to give feedback on

it) and Weibo (users report fake tweets and elite users scrutinize and judge them) are in function.

As the news articles usually contain a huge amount of text, this makes the input sequence long enough. In such cases, the old information gets washed off and scattered focus over the sequences which is due to a lack of explicit word alignment during decoding. Theirs raised a need to solve these issues and the attention mechanism has done a good job. There are different flavors of attention mechanisms that came up depending on their use cases, first and very basic version i.e., the basic attention which extracts important elements from a sequence. Multi-dimensional attention captures the various types of interactions between different terms. Hierarchical attention extracts globally and locally important information. Self-attention (*Vaswani et al., 2017*) captures the deep contextual information within a sequence. Memory-based attention discovers the latent dependencies. Task-specific attention captures the important information specified by the task.

Singhania et al. implemented a 3HAN hierarchical attention model (*Singhania, Fernandez & Rao, 2017*) that has three layers for words, sentences, and headlines each using bi-directional GRUs of the network encoding and attention. Wang et al. implemented Attention-based LSTM (*Wang et al., 2016*) for aspect-level sentiment analysis that finds the aspects and their polarity for the sentence. *Goldberg (2016)* applied a novel design for the NLP task that incorporates an attention-like mechanism in a Convolutional Network. Further enhancement with a deep attention model with RNN given by *Chen et al. (2018)* learns selective temporal hidden representations of the news item that bring together distinct features with a specific emphasis at the same time and generate hidden representation.

Convolutional Neural Networks were used in computer vision tasks but recently they have gained popularity in natural language processing tasks as well. CAMI (*Yu et al., 2017*) tries to early detect the misinformation. It is carried out by dividing the events into phases and representing them using a paragraph vector (*Le & Mikolov, 2014*). Automatic, identification of fake news based on geometric deep learning (Monti et al., 2019) generalizing classical CNNs to graphs. FNDNet (*Kaliyar et al., 2020*) deep convolutional neural network. DMFN (*Nguyen et al., 2019*) model capturing dependencies among random variables using a Markov random field. Pattern driven approach (*Zhou & Zafarani, 2019*) capturing the relation between news spreader and relation between the spreaders of that news item. A mutual evaluation model (*Ishida & Kuraya, 2018*) that dynamically builds a relational network model to identify credibility taking into consideration the consistency of the content. Without a dynamic relation network, the content dependent model would lead to a different score of the same article, since a different individual will have different perspectives. Several researchers have proposed various approaches to the detection of fake news as discussed in this section. Various classifiers and representation techniques are proposed. The reported accuracy for these models ranges from 85 to 90%. However, there is a scope for improving the accuracy of the fake news detection model.

From the above work, it can be observed that the a number of researchers have carried out a detailed study in finding out various linguistic features and using different combinations

of features in classification tasks. Deep learning automatically learns features from data, instead of adopting handcrafted features, it makes the task easier and can handle a huge amount of data. We have used the deep learning approach for feature extraction and added a new feature that helps improve the understanding of the text under consideration namely the stance which estimates the relative perspective of two pieces of texts. The major contribution over the previous works lies in the addition of stance as a feature along with the state of art BERT Model.

Recently BERT-based models are applied in NLP tasks, that is the hybrid of BERT and Artificial Intelligence techniques like RNN (*Kula, Choraś & Kozik, 2020*),CNN (*Kaliyar, Goswami & Narang, 2021*) and both (*Ding, Hu & Chang, 2020*). BERT models have also proved its importance to deal with multi-modal news articles (*Zhang et al., 2020*).

We have considered cosine distance between the vectors of the news title and the news body as the similarity measure. In traditional literature, the stance is defined in several ways. E.g., stance is detected towards a particular topic (*Sun et al., 2018*), agreement or disagreement towards a particular claim (*Mohtarami et al., 2018*) and even attitude expressed in a text towards a target (*Augenstein et al., 2016*). All of these have predefined categories like a negative, positive, neutral, agree, disagree, unrelated, for or against. Our intention here is to find the relation/similarity between the two text pieces(namely the title and the body of the text). Hence, we do not find the score towards a particular topic but the measure of similarity between the title and the body of the news. The reason we have made such a choice is that, for the unseen data, we are not already aware of the topic it is focusing on. This will make our system more generalized. Such an approach can identify how close or farther the title is to the text in the body of an article under consideration.

Due to this additional feature, our training data is better equipped for more accurate predictions. Also, with the help of state-of-art language model BERT, our model captures the semantics of the text efficiently with a multi-layer bidirectional transformer encoder which helps to learn deep bi-directional representations of text (article) and finetuning it on our training data to classify an article into fake or real, using the probability score our model assigns to it. In summary, the main contributions of this article are:

- Introducing stance as one of the features along with the content of the article to obtain state-of-the-art performance when detecting fake news using an AI model;
- Develop a model that captures the semantics of information using the pre-trained contextualized word embeddings BERT(Language Modelling Technique);
- Experimentation and validation of the above approach on the benchmark dataset.

The remaining article is structured as follows: 'Materials & Methods' outlines the methodology we have adopted to design our system, 'Experimental Setup' describes the experimental setup and parameters we are using, 'Results and Discussion' describes our findings and discussion on them, 'Conclusion' concludes our work.

## MATERIALS & METHODS

We are looking in this paper to build a classifier that detects fake news with better accuracy than already reported. We have experimented with multiple AI models and evaluated

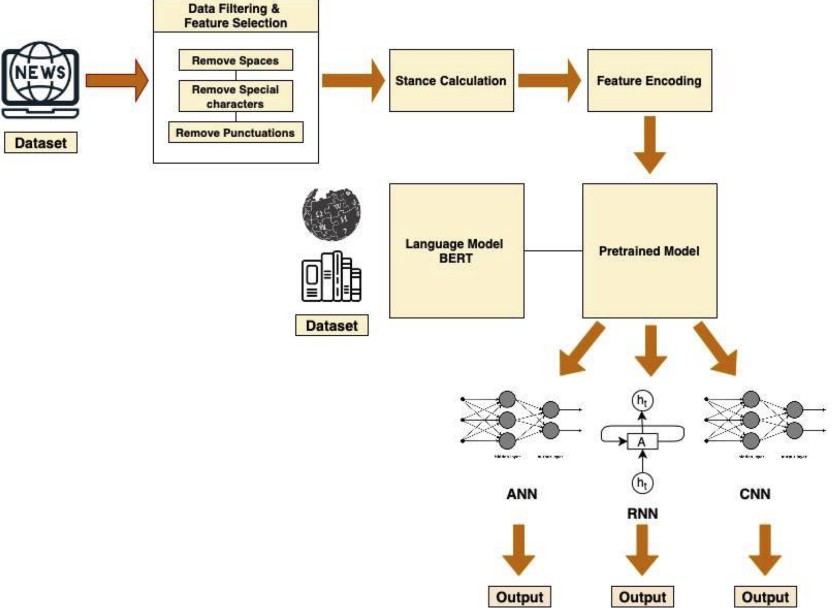

**Figure 1  Pipeline architecture of the system.**

accuracies. Our model for fake news detection is based on the content features which use pre-trained embedding to better capture the contextual characteristics. The complete process is explained in the below subsections.

## Pre-processing

It is observed that text data is not clean. The data is extracted from different sources hence they have different characteristics, to bring such data on a common ground text pre-processing is an important step. Our pre-processing pipeline depends on the embeddings we will be using for our task. Since most of the embeddings do not provide vector values for the special characters and punctuations we need to clean the data by removing such words. Our preprocessing step involves the removal of punctuations, special characters, extra spaces, and lowering the case of the text.

## Architecture

Figure 1 depicts the complete pipeline of our model.

## Dataset

Manual fact-checking is a tedious and lengthy task and such fact-checked articles are very less to train deep learning models. Researchers have extracted news articles from the websites they believe to be authentic as real or genuine and similarly the fake articles. McIntire Fake and Real News Dataset (https://github.com/joolsa/fake_real_news_dataset), a quite large dataset and a balanced dataset that contains both fake stories and true news. McIntire used the part from Kaggle's fake news collection for the Fake news class, while he got articles from credible journalism organizations like the New York Times, Wall Street

|   | title | text | label |
|---|-------|------|-------|
| 0 | You Can Smell Hillary's Fear | Daniel Greenfield, a Shillman Journalism Fello... | FAKE |
| 1 | Watch The Exact Moment Paul Ryan Committed Pol... | Google Pinterest Digg Linkedin Reddit Stumbleu... | FAKE |
| 2 | Kerry to go to Paris in gesture of sympathy | U.S. Secretary of State John F. Kerry said Mon... | REAL |
| 3 | Bernie supporters on Twitter erupt in anger ag... | — Kaydee King (@KaydeeKing) November 9, 2016 T... | FAKE |
| 4 | The Battle of New York: Why This Primary Matters | It's primary day in New York and front-runners... | REAL |
| 5 | Tehran, USA | \nI'm not an immigrant, but my grandparents ... | FAKE |
| 6 | Girl Horrified At What She Watches Boyfriend D... | Share This Baylee Luciani (left), Screenshot o... | FAKE |
| 7 | 'Britain's Schindler' Dies at 106 | A Czech stockbroker who saved more than 650 Je... | REAL |
| 8 | Fact check: Trump and Clinton at the 'commande... | Hillary Clinton and Donald Trump made some ina... | REAL |
| 9 | Iran reportedly makes new push for uranium con... | Iranian negotiators reportedly have made a las... | REAL |

**Figure 2** **A part of a dataset consisting of true and false news instances.**

Journal, Bloomberg, National Public Radio, and The Guardian for the real news class. The dataset contains 6335 news articles out of the 3171 that are genuine and 3164 that are fake. The dataset is equally distributed with ∼50–50% of real labeled articles and fake labeled articles (Refer Fig. 2). In real-world applications the data may be skewed, however, for experimentation and dataset building, we have considered a balanced dataset which leads to a more robust and generalized model.

## Data filtering and feature selection

The data filtering/cleaning is the task of preparing the data for encoding and training purposes. Here firstly we remove the spaces and punctuations from the text as they do not play an important role in the understanding of the context. This module of our pipeline ends up with feature selection for our training. We perform stance detection on the title and text to get the similarity score which gives us an additional feature to make our classifier perform better. We then encode our selected features.

## Stance calculation

We calculate the stance estimating the relative perspective of two pieces of the text concerning a subject, argument, or issue. It is the probability score that will be used as an additional feature to detect the misinformation. From Sensationalism detection by *Dong et al. (2019)* we consider that the similarity between the article body and article headline can be correlated with articles' credibility. The similarity is captured by first embedding the article body and article headline onto the same space and computing the cosine distance as the similarity measure. To achieve this we first tokenize the article's body and headline, then calculate the cosine distance between each article's body and headline pair. Cosine distance is a good similarity measure as it determines the document similarity irrespective of the size; this is because the cosine similarity captures the angle of the documents and not the magnitude. Mathematically, it measures the cosine of the angle between two vectors projected in a multi-dimensional space.

$$Cos\theta = \frac{\vec{a}.\vec{b}}{\|\vec{a}\|\,\|\vec{b}\|} = \frac{\sum_1^n a_i b_i}{\sqrt{\sum_1^n a_i^2}\cdot\sqrt{\sum_1^n b_i^2}}$$

Here, $\vec{a}.\vec{b} = \sum_1^n a_i b_i = a_1 b_1 + a_2 b_2 + \ldots + a_n b_n$ is the dot product of the two vectors.

## Feature encoding

Any AI model requires the input data to be in number format. Different state-of-art representation techniques like one-hot encoding, word embeddings like Continuous Bag of Words and Skip-gram, GloVe (Global vectors for word representation), Open AI GPT pre-training model internally using the transformer decoder concept, BERT (Bidirectional Encoder Representations from Transformers) bi-directional pre-training model that uses transformer encoder within are available to perform such encoding tasks. Our model uses language model BERT's smaller version namely Distil-BERT (*Sanh et al., 2019*) which has about half the total number of parameters of the BERT base. In Distil-BERT, a triple loss combining language modeling, distillation, and loss of cosine distance was included to exploit the inductive biases gained during training by these types of models. We are further fine-tuning the model for our text classification task on our dataset.

## Classification models

We have evaluated three AI models namely simple Artificial Neural Network, Recurrent Neural Network, Long Short-Term Memory (LSTM), Bidirectional LSTM, and Convolutional Neural Network.

1. Long Short-Term Memory (LSTM): LSTMs is chosen over RNN as it succeeds in overcoming the issue of long-term dependencies by keeping a memory at all times. The output of the embedding layer is fed into the LSTM layer that consists of 64 units and further passed through the dense layers with sigmoid activation and binary cross-entropy for computation. The next model implemented is the bidirectional LSTM model, which is an extension of traditional LSTM that trains two LSTMs in opposite directions to enhance model efficiency in capturing all the necessary information.

2. Bidirectional LSTM: Similar to the LSTM model it takes input from the embedding layers, our Bi-LSTM layer contains 64 units that are fed to the dense layers for computation. For comparison purposes, the activation and loss values used are the same as the previous model.

3. Convolutional Neural Network: Convolutional neural networks although designed for computer vision tasks recently it has given excellent results with NLP tasks as well. In the case of computer vision tasks, the filters slide over the patches of an image, whereas in NLP tasks the filter slides few words at a time over the sentence matrices. This makes Convolutional Neural Networks work well for classification tasks. So we have also implemented a CNN model that consists of a single Conv-1D with the kernel of size 5 and the Max Pooling layer. Further, a flattened layer and a fully connected layer with 64 nodes are used for computation. In all of the above models, we have used a Dense layer that operates linearly where every input is related by some weight to each output. The Loss function used is the cross-entropy loss that measures classification model performance whose output is a probability value between 0 and 1.Loss of cross-entropy increases as the predicted likelihood diverges from the actual mark. In the case of binary classification, where the

number of groups equals 2, cross-entropy can be determined as:

$$L = -(y\log(p) + (1-y)\log\log(1-p))$$

These models classify the news articles as fake or real using the sigmoid activation function.

## EXPERIMENTAL SETUP

The framework is developed in the Python backend and uses the python libraries namely Keras, NLTK, Pandas, Numpy, Sklearn, sci-kit, etc. The dataset was divided into training, validation and testing sets with train_test_split functionality of sci-kit learn. The scale of the training set size was 70 percent, the validation set scale of 15 percent, and 15 percent of the test set size. Data pre-processing involved the use of the NLTK tool for the removal of HTML tags, punctuations, multiple spaces, etc. The distil-Bert-base-uncased model from the hugging face is used to obtain the embeddings that are later fine-tuned on our dataset. We encode our data with the max length of 512, dimension to be 768, dropout of 0.1, and the number of layers as 6 to gain the input ids and the mask ids. For the classification purpose, we have used LSTM, Bidirectional, Conv1D, Dense layers from Keras. The number of units chosen for each of the layers was based on the experimentations carried out. For the GloVe model, different vector dimensions were tried 100 and 300 and the vector dimension 100 gave good accuracy results. The loss function and activation function used were cross-entropy loss and sigmoid activation for all the models.

### Performance metrics

We have used the Confusion matrix, Accuracy, Precision, Recall, F1, and ROC to evaluate our model's efficiency (*Hossin & Sulaiman, 2015*).

1. Confusion Matrix: The information performed by a classifier regarding actual and predicted classifications is studied by a confusion matrix.
2. Accuracy: Accuracy is the proportion of true outcomes within the total number of examined cases.
3. precision: Precision tells about what proportion of predicted Positives is truly Positive.
4. recall: It tells us what proportion of real positives is graded correctly.
5. F1 Score: It gives the harmonic mean of precision and recall.
6. ROC: ROC demonstrates how well the percentage of the positives are isolated from the negative groups.

These metrics helped us analyze the results we gained from our model. Table 1 depicts the values for accuracies during training, validation, and testing, along with the precision, recall, F1, and the ROC.

## RESULTS AND DISCUSSION

In this work, we have proposed an approach to upgrade the fake news identification system with the inclusion of an additional feature termed "stance". Stance helps us in understanding the relevance of the article title (i.e., the headline of the news) to the article body (i.e., the text of the news). We add this feature to our content features that are

**Table 1  Performance of different AI models.**

| | Models | Accuracy (%) | Precision (%) | Recall (%) | F1 (%) |
|---|---|---|---|---|---|
| Tokenizer | LSTM | 86.6 | 85.1 | 88.7 | 86.9 |
| | Bi-LSTM | 85.4 | 84.9 | 86.2 | 85.5 |
| | CNN | **93.0** | 93.0 | 93.0 | 93.01 |
| GloVe embeddings | LSTM | **92.1** | 91.7 | 92.7 | 92.2 |
| | Bi-LSTM | 91.9 | 90.2 | 93.9 | 92.1 |
| | CNN | 91 | 91.6 | 90.2 | 90.9 |
| GloVe embeddings and attention mechanism | LSTM | **92.1** | 91.7 | 92.7 | 92.2 |
| | Bi-LSTM | 91.9 | 90.2 | 93.9 | 92.1 |
| | CNN | 91 | 91.6 | 90.2 | 90.9 |
| BERT embeddings | LSTM | 91.16 | 91.01 | 91.01 | 91.01 |
| | Bi-LSTM | 93.05 | 88.76 | 88.76 | 93.3 |
| | CNN | **95.32** | 94.11 | 94.11 | 95.31 |

**Table 2  Classification results for the proposed model.**

| Models | Training accuracy (%) | Validation accuracy (%) | Testing accuracy (%) | Precision (%) | Recall (%) | F1 (%) | ROC (%) |
|---|---|---|---|---|---|---|---|
| ANN | 91.52 | 91.86 | 91.85 | 91.98 | 91.98 | 91.69 | 91.85 |
| LSTM | 98.51 | 91.16 | 91.16 | 91.01 | 91.01 | 91.01 | 91.15 |
| Bi-LSTM | 98.48 | 93.06 | 93.05 | 88.76 | 88.76 | 93.3 | 93.47 |
| CNN | 99.96 | 95.33 | **95.32** | 94.11 | 94.11 | 95.31 | 95.33 |

obtained from the pre-trained BERT model, which provides additional insight into the article. The AI models we have experimented with are ANN, LSTM, Bidirectional LSTM, and CNN. Results are obtained by training and testing these models with different vector representation techniques and even including the attention layer to some models. The results are presented in Table 1. Table 2 shows the classification results for our proposed model.

The best results we obtained are with the usage of a pre-trained language model. Table 1 shows the accuracies with different settings as with GloVe Embeddings we attain an accuracy of 92.1% for the LSTM model. With the inclusion of the Attention layer to these models the accuracy shows some improvement of about 1%. Our proposed model adapting the BERT embedding to capture the contextual information from the articles have proved to perform better with an accuracy of 95.32%. We have experimented and obtained the results using the pre-trained BERT model and fine-tuned BERT model. The results we obtained via both these models demonstrate a negligible difference. One of the possible explanations for this could be that the BERT is trained on the English language corpus. The dataset we have used for experimentation is also in English and has a similar structure and features.

To show the effect of stance as a feature we have experimented with BERT encoding, which builds a representation of the news title and news body along with the similarity

**Table 3 Effectiveness of stance feature in the classification of news articles.**

| Features | Models | Training accuracy (%) | Validation accuracy (%) | Testing accuracy (%) | Precision (%) | Recall (%) | F1 (%) | ROC (%) |
|---|---|---|---|---|---|---|---|---|
| News Title, News Body | ANN | 89.2 | 88.0 | 88.33 | 86.57 | 86.57 | 88.61 | 88.41 |
| | LSTM | 95.29 | 88.8 | 90.64 | 87.42 | 87.42 | 91.0 | 90.9 |
| | Bi-LSTM | 97.99 | 89.79 | 92.21 | 93.6 | 93.6 | 92.0 | 92.26 |
| | CNN | 99.1 | 93.68 | 93.90 | 91.3 | 91.3 | 94.0 | 94.0 |
| News Title, News Body, Similarity between them (Stance) | ANN | 89.31 | 89.37 | 89.38 | 86.4 | 86.4 | 89.8 | 89.4 |
| | LSTM | 94.36 | 89.05 | 91.06 | 87.8 | 87.8 | 91.44 | 91.37 |
| | Bi-LSTM | 98.6 | 92.11 | 92.6 | 93.5 | 93.5 | 92.5 | 92.6 |
| | CNN | 99.3 | 92.9 | 94.42 | 94.33 | 94.33 | 94.43 | 94.42 |

between them, and demonstrated that it outperforms encoding the news title and news body alone. From the results, it can be observed that there is a considerable increase in the testing accuracy (Refer Table 3).

We have dealt only with the content part of the article, the reason being when a news article is published and not much-circulated yet, the metadata such as reposts, likes, shares, etc. are not available. Then content can be the only parameter considered for fake news detection. The below plots give a clear view of the results obtained from our model (Refer Fig. 3). We have carried out a 5-fold Cross-validation resampling procedure to evaluate our model and make the results comparable with the other models on the same dataset (Refer Table 4). We implemented a stratified k-fold cross validation, however observe a few misclassified samples in the test results. This is primarily due to the overlapping of features in the two classes and having unclear distinction due to that.

To validate our approach, we carried out a comparison with pre-existing approaches by considering a benchmark dataset by G. McIntire Fake and Real News Dataset. Many have proposed a solution to the fake news detection problem using hand-crafted feature engineering and applying Machine Learning models like Naïve Bayes, Random Forest, Boosting, Support Vector Machine, Decision Tree, Logistic Regression. *Reddy et al. (2020)* reported an accuracy of 95% with a gradient boosting algorithm on the combination of stylometric and CBOW (Word2Vec) features. *Bali et al. (2019)* used Sentiment polarity, Readability, Count of words, Word Embedding, and Cosine similarity as the features to discriminate fake news with machine learning models. He reported the highest accuracy of 87.3% with the XGBoost model. *Esmaeilzadeh, Peh & Xu (2019)* uses the LSTM-encoder–decoder with attention to the text, headline, and self-generated summary of the text to obtain the features. He proved that the summary of the text gives better accuracy of 93.1%. Embedding methods such as LSTM, depth LSTM, LIWC CNN, and N-gram CNN were used (*Huang & Chen, 2020*) and weights were optimized using the Self-Adaptive Harmony Search (SAHS) algorithm with an initial accuracy of 87.7% that was increased to 91%. *Khan et al. (2019)* in his research he used word count, average word length, article length, count of numbers, count of parts of speech(adjective), count of exclamation along with the sentiment feature, and n-gram GloVe embedding encoded features. His work

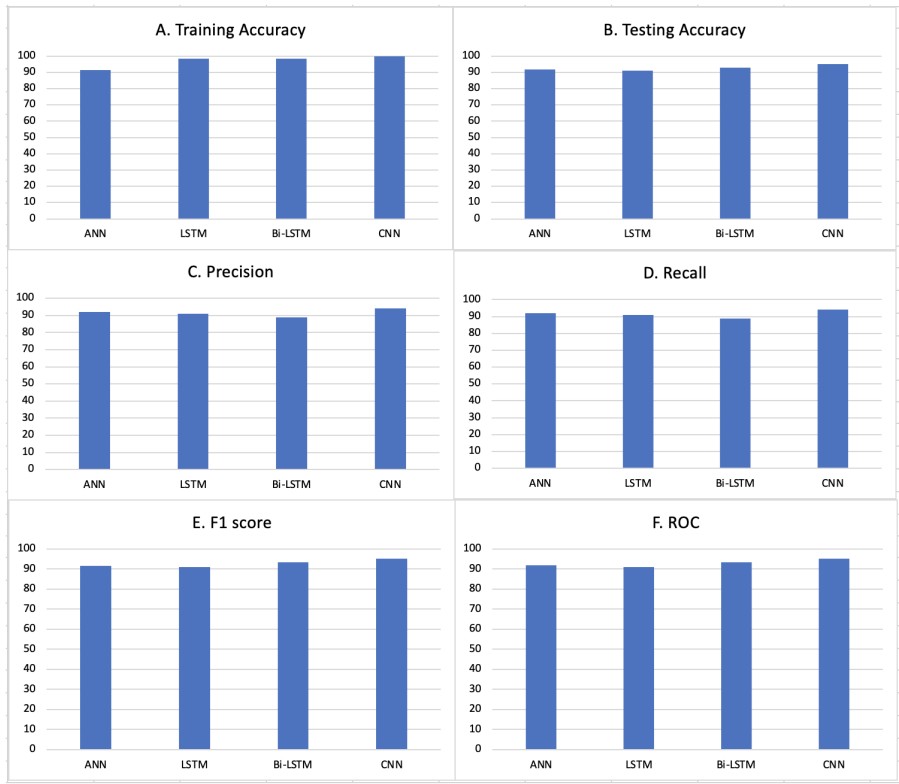

**Figure 3  Different evaluation metrics applied to our system.** (A) Training accuracies for different models. (B) Testing accuracies for different models. (C) Precision scores for different models. (D) Recall values for different models. (E) F1 Scores for different models. (F) ROC values for different models.

reported accuracy of 90% with the Naïve Bayes classifier and 95% with the LSTM deep neural network. *Bhutani et al. (2019)* reported an accuracy of 84.3% with the Naïve Bayes model and TF-IDF scores, sentiments, and Cosine similarity scores as features. *Gravanis et al. (2019)* developed a content-based model and reported an accuracy of 89% with the Support Vector Machine. *George, Skariah & Xavier (2020)* in his work used CNN for Linguistic features and Multi Headed self-Attention for the contextual features. Accuracies of 84.3% and 83.9% were achieved with machine learning models such as the Naïve Bayes classifier and Random forest respectively. With deep learning models like LSTM and FastText embedding accuracy reported was 94.3%. With the increasing creation and availability of misinformation, automatic feature extraction with the help of Deep Learning algorithms has also been experimented.  A comparison of other work with our model is shown in Table 5.

## CONCLUSION

Information disorder and the spread of fake news is a highly prevalent and challenging problem in this digital era. The highly harmful potential of fake news on social stability and the psychological health of the masses is undisputed and proven time and again. To identify

**Table 4 5-fold cross validation results for the proposed model.**

| Features | Models | Training accuracy (%) | Validation accuracy (%) | Testing accuracy (%) | Precision (%) | Recall (%) | F1 (%) | ROC (%) |
|---|---|---|---|---|---|---|---|---|
| News Title, News Body, Similarity between them (Stance) | ANN | 91.31 | 88.72 | 90.73 | 87.35 | 87.35 | 91.14 | 91.17 |
| | LSTM | 97.08 | 87.61 | 88.60 | 84.98 | 84.98 | 89.29 | 89.40 |
| | Bi-LSTM | 99.23 | 92.72 | 93.24 | 92.03 | 92.03 | 93.34 | 93.44 |
| | CNN | 99.92 | 95.25 | 95.85 | 94.81 | 94.81 | 95.89 | 95.90 |

**Table 5 Performance of various Fake News Identification models.**

| Models | NB | RF | SVM | MLP | Boost | LSTM | Bi-LSTM | CNN |
|---|---|---|---|---|---|---|---|---|
| *Reddy et al. (2020)* | 86 | 82.5 | | | 95 | | | |
| *Bali et al. (2019)* | 73.2 | 82.6 | 62 | 72.8 | 87.3 | | | |
| *Bhutani et al. (2019)* | 84.3 | 83.9 | | | | | | |
| *Gravanis et al. (2019)* | 70 | | 89 | | | | | |
| *Bharadwaj & Shao (2019)* | 90.7 | 94.7 | | | | 92.7 | | |
| *George, Skariah & Xavier (2020)* | 84.3 | 83.9 | | | | 94.3 | | |
| *Esmaeilzadeh, Peh & Xu (2019)* | | | | | | 92.1 | 93.1 | |
| *Huang & Chen (2020)* | | | | | | 84.9 | 87.7 | 91 |
| *Khan et al. (2019)* Char-LSTM | 90 | | | | | 95 | 85 | 86 |
| Our Model | | | | | | 91.16 | 93.05 | **95.32** |

and detect the misinformation is the primary step in controlling its spread and combat its harmful effects on society. To this end, in this paper, we have proposed a methodology to identify the fake news based on stance detection in addition to other standard features. The content of an article is the basic informative piece that does play a significant role in assigning credibility to the article. We propose a model that uses language features based on the article's content to discriminate against fake news from real news. Our model tries to detect the fake articles at a stage when they are yet to propagate in the social network. To make the detection more precise we have added a stance feature to the dataset. This stance value has helped us understand the content by finding the relevance between the article headline and article body. Along with this additional feature, we have learned the language representation with the help of the BERT technique. The transfer learning concept that injects external knowledge gives a better understanding of our content. It is observed that our system has shown improvement in the results as compared to the other models. Previous work is conducted with handcrafted linguistic features, stylometric features, TF-IDF scores, n-gram features as well as automatic feature extraction from the content of the articles. Evaluation of our model is done on Kaggle's open source dataset for news articles. The results presented demonstrate the performance of various deep learning architectures for misinformation detection, wherein the BERT embeddings based CNN architecture provides the highest performance of 95% accuracy and comparable precision and recall. Besides, we have also compared our approach with other pre-existing approaches and shown that our proposed model provides superior performance as compared to the other methods. One of the limitations is that it is essential to have required features in the

dataset. Without this, this approach will not work effectively. We will extend this work in the future to overcome this limitation.

On the social media platforms, images, videos and audio have become a part of a news article to make it more informative and sensational at times. The fake news articles are usually intertwined with the factual information that makes interpretation of the contextual information difficult. As an extension of this work image, video, and audio can also be used in the content-based detection systems to make them near to perfect.

### Funding
This work was supported by the Scheme for Promotion of Academic and Research Collaboration (SPARC) 2018-19, MHRD (project no. P571). The funders had no role in study design, data collection and analysis, decision to publish, or preparation of the manuscript.

### Grant Disclosures
The following grant information was disclosed by the authors:
Scheme for Promotion of Academic and Research Collaboration (SPARC) 2018-19.
MHRD: P571.

### Competing Interests
Ketan Kotecha is an Academic Editor for PeerJ. The authors declare there are no competing interests.

### Author Contributions
- Hema Karande conceived and designed the experiments, performed the experiments, analyzed the data, performed the computation work, prepared figures and/or tables, authored or reviewed drafts of the paper, and approved the final draft.
- Rahee Walambe conceived and designed the experiments, analyzed the data, prepared figures and/or tables, authored or reviewed drafts of the paper, and approved the final draft.
- Victor Benjamin and T.S. Raghu analyzed the data, authored or reviewed drafts of the paper, and approved the final draft.
- Ketan Kotecha conceived and designed the experiments, analyzed the data, authored or reviewed drafts of the paper, and approved the final draft.

### Data Availability
The dataset and code are available at GitHub: https://github.com/hemakarande/Fake-News-Detection/tree/master.

### Supplemental Information
Supplemental information for this article can be found online at http://dx.doi.org/10.7717/peerj-cs.467#supplemental-information.

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
