# Peer review of "Stance detection with BERT embeddings for credibility analysis of information on social media"

_PeerJ Computer Science, doi:10.7717/peerj-cs.467_

## Round 0.1 · original submission · Major Revisions

Having received reviews from three reviewers, they have all made comments suggesting substantial amendments in the submission before it can be considered for publication. While some of the concerns are linked to strengthening the presentation and analysis of the results, some other concerns are more substantial and should be addressed carefully.

There are concerns about the use of a similarity metric to compute the stance of a news article, which would need further justification if it's used as is now. There are also concerns about the paper not actually tackling the stance detection task, but rather a clickbait detection task, which is perhaps more sensible to be tackled with the similarity metric. I would suggest formally defining the task and justifying the use of appropriate methods.

Use of a single dataset and lack of methods for cross-validation limit the generalisability of the results as well as their comparability with respect to previous work. Making experiments comparable with previous methods and use of additional datasets would provide clear improvements to strengthen the contribution of the paper.

Please see detailed comments from reviewers suggesting revisions in the aforementioned and other directions, which should be useful in preparing a major revision.

Reviewer 1 ·

Basic reporting

The authors propose investigate the performance of LSTM, bi-LSTM and CNN neural models with BERT embeddings and stance feature for credibility analysis on text resources.

The paper is well structured and easy to follow. The challenged problem and the motivation to study this problem are described clearly. However the introduction is too lengthy describing the social aspect of the problem. Previous solutions in the literature are described well, however the difference of the proposed method and contribution over the previous work are not given clearly.

The figures and the charts are structured appropriately.

Experimental design

The authors focus on the use of BERT embeddings and the similarity between the title and the content of the news text for credibility analysis. In a vast number of recent studies, we see that BERT has been used for different NLP related problems. So using BERT in credibility analysis is inline with the recent efforts. However the contribution is limited since the BERT pre-trained model is directly used without any further training effort.

The use of similarity between the title and the content of the news text is interesting. However there are several drawbacks about the definition and the analysis of the feature. First of all, this new feature is considered to be about stance of the news, however the measured similarity may not be about the stance. Since the title and body texts are represented through the terms, the similarity may be related with the topic. As another drawback at this point, the details as to how the vector is constructed are not given. As another limitation, it is not clear why this feature is considered and how effective it is. There is not experiment in the study which is designed to measure the effect of this feature particularly on the credibility analysis. All the experiments involve the BERT embeddings and this feature together.

Another limitation of the study is that a single data set is considered. With a quick search, one can find various data sets on credibility analysis.

Validity of the findings

The results are not obtained under cross validation. Therefore they are open to bias in partitioning of the data set for training, test and validation.

The authors present a performance comparison against the results reported on the same data set, however it is not clear how comparable they are due to lack of cross validation and also since the data partitioning used in the results reported in other studies may be different.

Additional comments

The authors challenge a contemporary problem and present the study clearly. The motivation and the social aspect of the problem are discussed a bit lengthy but It is an enjoyable manuscript to read.

Using BERT embedding is a nice approach following recent developments in the area. Using title/body text similarity is a good idea with potential. However the contribution of the feature to the credibility analysis is not analyzed well.

Unfortunately there are other limitations with the experimental setup. Since the experiments are not conducted with cross validation it is not clear how comparable the results of different settings and the results reported on previous work. Use of just a single data set is also a limitation to generalize the results.

Reviewer 2 ·

Basic reporting

no comment

Experimental design

no comment

Validity of the findings

no comment

Additional comments

The authors have proposed an approach to detect fake news using BERT embeddings and reported high accuracy scores. However, there are some issues in the paper that need to be addressed:

- The major issue is the use of the terminology ‘stance detection’. What the authors have done is not stance detection; at best it could be seen as a naive approach for clickbait detection. According to [1], “Stance detection aims to assign a stance label (for or against) to a post toward a specific target.” For more related works refer to [2], [3].

- The title is slightly misleading. Credibility analysis and fake news detection are two related but different tasks. The title needs to be modified to make readers aware of what to expect in the paper.

- “Misinformation or Information Disorder in the form of fake news is usually in the form of completely false information, photographs or videos that are intentionally created and distributed to confuse or misinform…” - Firstly, this is not exactly true. Disinformation and fake news may not be completely false information. In fact, in most cases, these are statements, claims and narratives with some degree of truth, taken out of context to mislead and deceive readers. Images and videos may not be intentionally created for this purpose - again, genuine images may simply be taken out of context to deceive viewers.
Secondly, the structure of this sentence needs to be changed and made more understandable.

- The division of sections and subsections is very confusing and this needs to be fixed: The ‘Previous Work’’ section is presented as a subsection of the ‘Social and Cognitive Aspects’. I would suggest removing the ‘Social and Cognitive Aspects’ section, since there is a huge body of literature that needs to be explored and explained, if this section is to be included somewhere later on in the manuscript.

- The architecture of the model needs to be described in words as well, alongside the diagram.

- Connections need to be made to the pre-existing works in the credibility literature.

- Change all occurrences of ‘dis-information’ to ‘disinformation’.

- URLs in footnotes 4 to 13 can be shortened, or moved to the references.

- The manuscript needs to be thoroughly proofread to remove grammatical errors.




References:

[1] Sun, Q., Wang, Z., Zhu, Q., & Zhou, G. (2018, August). Stance detection with hierarchical attention network. In Proceedings of the 27th international conference on computational linguistics (pp. 2399-2409).

[2] Augenstein, I., Rocktäschel, T., Vlachos, A., & Bontcheva, K. (2016). Stance detection with bidirectional conditional encoding. arXiv preprint arXiv:1606.05464.

[3] Mohtarami, M., Baly, R., Glass, J., Nakov, P., Màrquez, L., & Moschitti, A. (2018). Automatic stance detection using end-to-end memory networks. arXiv preprint arXiv:1804.07581.

Reviewer 3 ·

Basic reporting

no comment

Experimental design

(1) Please state whether the parameters of the BERT model used in training was fine-tuned or just fixed.
(2) Regarding the class label ratio of the used dataset, to better simulate the situations in real-world applications (fake news are few), it would be better to conduct experiments trying different settings of the class-label-ratio, such as 1:5 for moderately skewed label distribution and 1:10 for high skewed distribution. This can be done by experimenting with a single model (e.g., the best one).
(3) As the stance signal is crucial to the task, it would be informative to show the differences between the stance scores (cosin) of fake news and those of the true news in the dataset.

Validity of the findings

(1) It would be worthing doing a statistical test on the results, in order to show that the differences between the results of the baseline methods with the proposed method are indeed statistically significant. To be able to do this, the authors may consider applying k-fold cross-validation to the dataset so that it is possible to obtain multiple results from each baseline and the proposed method for performing the statistical test.

(2) It would be also worthing to do an error analysis to discuss and compare the errors made by the baseline methods and the proposed method, e.g., showing test examples that are misclassified by the proposed method and speculate why the errors may occur.

---

## Round 0.2 · Minor Revisions

The paper has improved substantially from its previous revision, as acknowledged by the reviewers. There are however some final revisions required prior to acceptance:

- Discuss recent literature using BERT-based models (see references suggested by one of the reviewers).

- Perform error analysis to provide more detailed insights into the results.

- Either perform experiments on an additional dataset or acknowledge the limitations of experimenting on a single dataset.

Reviewer 1 ·

Basic reporting

The revisions in the organization and the writing are satisfactory.

Experimental design

The authors responded to my earlier comments satisfactorily for most of the items. Especially the new experiments under cross validation and analyzing the effect of stance improved the analysis part.

The only comment that have not been attempted is the experiments with additional data sets. The authors consider the results reported on Kaggle's Fake News data set and make a comparison over this data set. However ISOT's Fake news Dataset is also used in the previous studies.

https://www.uvic.ca/engineering/ece/isot/datasets/fake-news/index.php

Analysis on an additional data set is crucial to show that the success of the proposed method is not specific to a certain case. So I'd suggest including analysis and comparison on this data set as well.

Validity of the findings

There are several recent studies on BERT based fake news detection.

Some of them are as follows:

Zhang, Tong, et al. "BDANN: BERT-Based Domain Adaptation Neural Network for Multi-Modal Fake News Detection." 2020 International Joint Conference on Neural Networks (IJCNN). IEEE, 2020.

Kula, Sebastian, Michał Choraś, and Rafał Kozik. "Application of the BERT-Based Architecture in Fake News Detection." Conference on Complex, Intelligent, and Software Intensive Systems. Springer, Cham, 2020.

Kaliyar, Rohit Kumar, Anurag Goswami, and Pratik Narang. "FakeBERT: Fake news detection in social media with a BERT-based deep learning approach." Multimedia Tools and Applications (2021): 1-24.

Ding, Jia, Yongjun Hu, and Huiyou Chang. "BERT-Based Mental Model, a Better Fake News Detector." Proceedings of the 2020 6th International Conference on Computing and Artificial Intelligence. 2020.

In order to present the validity of findings in the paper, I'd suggest summarization such BERT related recent approaches and if possible comparing the reported results.

Additional comments

Hence I'd suggest

- including analysis on ISOT fake news data set

- including comparison with recent BERT based fake news detection approaches in the literature.

Reviewer 3 ·

Basic reporting

N/A

Experimental design

The authors have largely addressed the comments regarding experimental design, by showing the results using both pre-trained and fine-tuned models and effects of different features used. However, there is still a lack of results on comparing settings with different class-label-ratios, as suggested in the second comment of the "Experimental design".

Validity of the findings

The authors have nicely addressed the comment regarding the statistics test. However, the second comment on an error analysis on the testing results is still missing.

Additional comments

I would suggest the author do an error analysis on the testing results (e.g. a discussion of three misclassified test examples), which can help undertand where and how the proposed method fails.

---

## Round 0.3 · accepted · Accept

I have checked the authors' response and the new revision, and the authors have added the suggested references and acknowledged limitations as suggested by the reviewers and editor in the previous round. Therefore the submission can now be accepted in its current form.